# An autonomous organic reaction search engine for chemical reactivity

Vincenza Dragone[1], Victor Sans[1], Alon B. Henson[1], Jaroslaw M. Granda[1] & Leroy Cronin[1]

The exploration of chemical space for new reactivity, reactions and molecules is limited by the need for separate work-up-separation steps searching for molecules rather than reactivity. Herein we present a system that can autonomously evaluate chemical reactivity within a network of 64 possible reaction combinations and aims for new reactivity, rather than a predefined set of targets. The robotic system combines chemical handling, in-line spectroscopy and real-time feedback and analysis with an algorithm that is able to distinguish and select the most reactive pathways, generating a reaction selection index (RSI) without need for separate work-up or purification steps. This allows the automatic navigation of a chemical network, leading to previously unreported molecules while needing only to do a fraction of the total possible reactions without any prior knowledge of the chemistry. We show the RSI correlates with reactivity and is able to search chemical space using the most reactive pathways.

[1] WestCHEM, School of Chemistry, University of Glasgow, Glasgow G12 8QQ, UK. Correspondence and requests for materials should be addressed to L.C. (email: Lee.Cronin@glasgow.ac.uk).

The number of small organic molecules reported in the literature is >75 million[1], yet this only represents an infinitesimal fraction of the estimated $10^{60}$ molecules available to search in chemical space[2,3]. To help speed up the process of chemical synthesis, numerous time-saving devices have been developed for today's modern chemical laboratory, including flow systems[4–6], automated chromatography columns, in-line analysis[7,8] and combinatorial screening that uses spectrometry[9]. Also, the development of flow systems for the synthesis of known molecules is well established and is particularly useful for automating chemical processes[10], as well as for developing safer procedures for working under more extreme conditions, or with highly reactive materials[11–13]. In addition, the availability of configurable robotics (for example, three-dimensional-printer-based chemical robots)[14], as well as a large variety of control systems[15–17], have been introduced for custom chemistry, target syntheses[18–20] and for optimization purposes[21–28]. Automation offers advantages such as reliable control over a chemical process increasing reproducibility[29] and has been used to enable the development of systems that allow the automatic optimization of the reaction conditions for known reactions[30–32]. Despite these advances, the autonomous 'closed-loop' searching of organic[33,34] chemical space following chemical reactivity[33] has not yet been achieved, which means the discovery of new reactions and molecules is still largely in the domain of the chemist.

To date, the development of automated systems in chemistry[35] has focussed on the generation of predesigned libraries, while the real-time experimental exploration of chemical reactivity remains underexplored. This is because the process of organic synthesis is normally target-based, and reactivity searches are normally focussed on a particular transformation/optimization of a given reaction step[24–27]. We therefore hypothesized that the 'closed-loop' exploration of chemical space could be implemented using a simple approach using spectroscopic feedback focussing on the differences between the starting reagents and the products enabling the search for new molecules, reactions and synthetic pathways. This is because the ability to search for reactivity, rather than following the constraints of a design-to-target approach might lead to the discovery of new reactions and molecules by following reactivity first. This approach could allow the elimination of bias which can prevent the human experimenter from doing a particular set of experiments. By focussing on a new metric-based approach following chemical change, the system will be able to explore without bias thereby allowing machine learning based upon only sensor feedback rather than relying on prior knowledge. Ideally, a system that could allow both searching of new chemical space and the ability to update the database to predict new routes would be the most powerful combination[36].

Herein we present a reactivity explorer robot that is capable of navigating a large reaction network and assessing the reactivity of the chemical transformations autonomously, without needing to do every reaction. The navigation of the chemical network is achieved in real time with an algorithm that compares the differences between the starting materials and each reaction mixture generated with no prior knowledge of the chemistry and without any work-up. The robot then uses the same algorithm to autonomously assess and rank the reactivity of all reagent combinations (represented as potential reaction pathways) quantified with a reaction selection index (RSI) and select the most reactive pathway automatically. The idea is not just to follow yield but to see how using a straightforward algorithm, following maximum reactivity, can lead to linking together many reactions with minimum work-up allowing the system to autonomously follow reactivity. In this instance, we chose a

model reaction network that comprises 64 unique possible reaction pathways arising from the combination of 12 reagents in groups of three ($4 \times 4 \times 4$); that is, a three-step synthesis where each step has a choice from one of the four possible reagents. During each run, the reagents travel through each of the three reaction manifolds, an infrared spectrum is collected using an in-line attenuated total reflectance infrared (ATR-IR) flow cell and then compared in real time with that of the starting materials in order to derive the RSI. After each reaction step, the combination with the largest RSI value is selected automatically, reacted in subsequent reaction steps and analysed in a similar manner. In this way, the decision-making system navigates through the reaction network following the reactions with the highest reactivity and hence is able to autonomously pursue the most reactive pathways under any given conditions.

## Results

**The reaction network/the model system.** The reaction network presented in our study is defined by a core molecule that is able to undergo three consecutive reactions: (I) Diels–Alder reaction; (II) reductive amination; and (III) amide formation. These three reactions were chosen as a proof of concept in order to gradually increase the complexity of the mixtures obtained in each step and obtain new unpublished molecules demonstrating the potential aiming for reactivity rather than new molecular identities. To achieve this, we selected a core 'framework' molecule that can be combined with one of the four available reagents in each reaction step as shown in Fig. 1a. However, using the RSI approach, the navigation of the reaction network is only focussed on the most reactive pathways. Indeed, at the end of each reaction step a decision point is reached, where a pathway is selected and used to direct the chemical navigation of the network, see Fig. 1b. In each generation, one of the four possible reagents was selected such that it should purposely not react under the selected conditions. These so called 'stopper' reagents were selected not only to act as an in-built control, as they should always score the smallest RSI, but they were also introduced to the system to investigate whether the algorithm would be capable of correctly assessing and differentiating between kinds of reactivities in real time.

The core molecule chosen was the diene 1,3-cyclopentadiene (**1**), which can then be reacted in the first-generation reaction with one of four reagents including three dienophiles—methacrolein (**2a**), trans-2-methyl-pent-2-enal (**2b**) and trans-2-pentenal (**2c**)—in order to form [4 + 2] cycloadducts (**3**) as the first-generation products. These also have an aldehyde moiety that can take part in the subsequent step by reacting one of the four reagents including three primary amines—aniline (**4a**), S-( − )-α-methylbenzylamine (**4b**) and cyclohexylamine (**4c**)—to form imines (**5**) to form the second-generation products. Next, in the presence of a reducing agent, the reaction between these cycloadducts (**3**) and the primary amines (**4**) yields secondary amines (**6**), which can then be reacted in the final step with one of the four reagents, three of which are the acyl chlorides—propionyl chloride (**7a**), benzoyl chloride (**7b**) and phenyl acetyl chloride (**7c**)—to form amides (compounds **8**) to yield the third generation as the final products of the three-step reaction sequence. The stoppers that were selected for the three generations were cyclohexanone (**2d**), N,N-dimethylaniline (**4d**) and methyl benzoate (**7d**) for generations I, II and III, respectively, see Fig. 2.

**The setup and the algorithm.** The physical robot reactor setup comprises of 14 digitally controlled 5 ml syringe pumps, three reactor manifolds: **R1**, **R2** and **R3** with volume of 1.5, 1.5 and 3 ml, respectively, and two in-line analytic instruments:

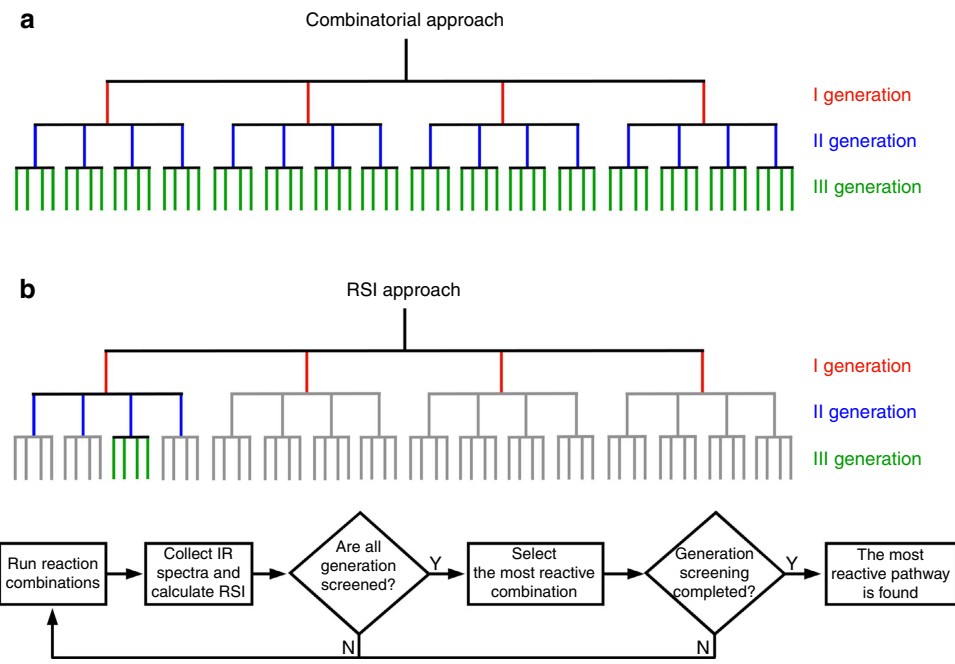

**Figure 1 | Difference between the combinatorial approach and the RSI approach for the investigation/navigation of the reaction network.**
(**a**) Schematic representaion of a 4 × 4 × 4 reaction network obtained using a combinatorial approach. A core molecule (donted by the black line) is combined with four reactants in three consecutive reaction steps. This means that it is possible to form a I, II and III generation of products consisting of 4, 16 and 64 reaction outcomes, respectively. (**b**) Top: schematic representation of the RSI approach that uses this index to direct the reaction network navigation to the most reactive pathways. Bottom: flow diagram of the algorithm used by the presented decision-making platform, for the navigation through different paths of the reaction network.

an ATR-IR and an electrospray ionization mass spectrometry (ESI-MS) spectrometers, see Fig. 3. The software to control the platform was written using LabVIEW. The software was able to fully automate and synchronize the system during each operation, in-line data acquisition and real-time data treatment and decision-making with the algorithm.

The core feature of the robot is the algorithm used to assess reactivity of the regents mixed together in the reaction manifolds with the RSI, which is derived from the mean square error (MSE) of the difference of the ATR-IR spectra. Once the run of a generation is completed, the RSI of each of the pathways screened is calculated as shown in equation 1 (see Supplementary Equations 1–5 for a full explanation):

$$\mathrm{RSI}_n = \frac{\mathrm{MSE}_n}{\sum \mathrm{MSE}_{\mathrm{gen}}} \qquad (1)$$

Here the MSE gives an estimate the difference between the infrared spectra of a known combination of starting materials and the same reacted (under specific conditions). This value is defined as the mean of the squares of the difference between the actual observations—in this case, an experimental infrared spectrum—and those predicted—in this case, the sum of the starting material infrared spectra (See Supplementary Equations 1 and 2).

**Assessing reactivity under different reaction conditions.** In initial experiments, we investigated whether the RSI in our algorithm can be helpful in assessing the reactivity of a reaction repeated under different conditions. We began with the study of the Diels–Alder reaction between compounds **1** and **2a** under nine different reaction conditions (see Table 1). These experiments were comprised of reactions at three different temperatures at three different residence times. In all cases, the conversion rates could be exactly calculated by integration of the aldehyde peak from the starting material, **2a**, and that of the cycloadduct, **3a**,

after curve-fitting operations. These results were then confirmed by [1]H-NMR spectroscopy. The results obtained in terms of conversion, RSI and a selected region (between 1750 and 1650 cm[−1]) of the infrared spectra of this reaction under the nine different conditions are summarized in Table 1. From these comparisons, it is possible to observe the highest conversion rate associated with a highest aldehyde intensity **3a**: **2a** ratio (corresponding to condition IX and to the red spectrum in graph **c** of Table 1). It can be seen that, for this reaction, the RSI values can be reliably utilized for the selection of the reaction condition giving the highest conversion.

**Multi-step reaction network navigation.** Now we have established that the RSI could be used to link reactivity with conversion, we set out to design and perform experiments intended to challenge the algorithm by making the analysis of reaction mixtures progressively complex. Indeed, in initial experiments, we investigated whether the RSI in our algorithm can reliably navigate a reaction network made of only two-step reaction pathways characterized by high atom economy: a Diels–Alder reaction and imine formation and carried on in the same tetrahydrofuran solvent media. Adopting this approach, the programmed set of experiments is automatically executed where the whole of generation I is run and a fourth of the generation II in 280 min. The screening of the generation I consisted of reacting compound **1** with the dienes **2a–d**. In the first step, the experiment concluded with the selection of the reaction pathway associated with the formation of compound **3a** as the first decision point. Interestingly, the stopper did not merely score the smallest RSI (as expected), but this value was an order of magnitude smaller than the other three (see Supplementary Table 17). Also, the RSI assessment of the generation I reaction pathways correctly matched the at-line [1]H-NMR analysis of the reaction mixtures, used to calculate the conversion yields of each

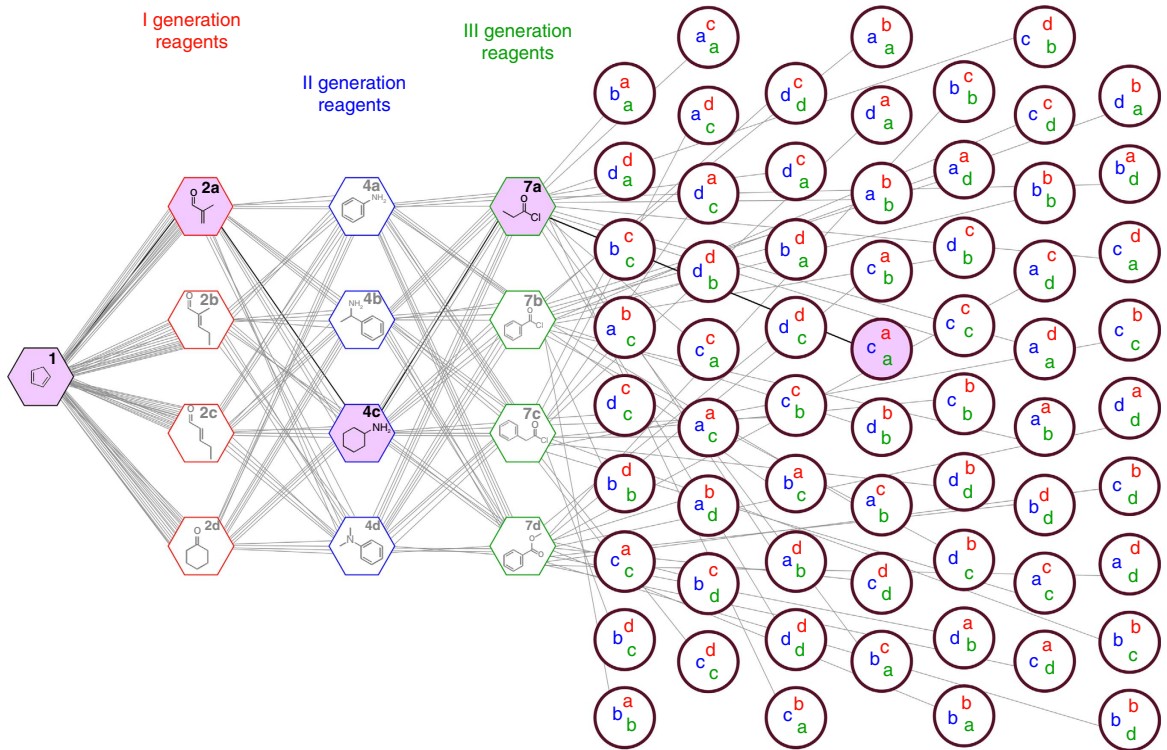

**Figure 2 | The reaction network used as chemical model system for the development of the autonomous organic search engine for chemical reactivity.**
Illustration of the reaction network obtained by reacting a core molecule (hexagon in black) in three-step reactions with four different reagents in each step.
The I, II and III generation reagents are represented by the hexagons in red, blue and green, respectively. The expected products of reactions are
represented with circles. The final selected pathway and reaction mixture are highlighted in purple.

of the diene **1** into the expected cycloadduct **3**. The navigation
then continued with the screening of generation II reactions
obtained from the combination of compound **3a** with **4a–d**. After
the screening of the four II generation reactions, the pathway
corresponding to the reaction between **3a** and **4c** was selected,
in which the aldehyde **3a** is converted to the imine **5c**. Again, the
stopper resulted in a value that is an order of magnitude smaller
than the selected pathway, and the selected pathway was shown to
be the most reactive by at-line [1]H-NMR analysis of these four
reaction mixtures (see Supplementary Figs 6–13).

The second experiment consisted of navigating the whole
reaction network as shown in Fig. 4a. This includes the increased
complexity of the reaction mixtures analysed, not only due to the
low atom economy of the reactions but also due to the use of
solvent mixtures. The screening of first generation concluded
in 120 min with the selection of the pathway forming **3a**
as previously described. The four experiments in the second-
generation reactions were conducted in the presence of a reducing
agent (to activate the imine for the subsequent step). These
experiments were concluded in only 160 min with the selection of
the reaction pathway corresponding to the formation of the
reaction mixture **6c**. The navigation continued with the screening
of the reactions from the third generation obtained from the
combination (with a molar and volumetric ratio 1:1) of
compound **6c** with **7a–d**. After these four reactions were ran,
the experiment was complete and the system selected compound
**8a** corresponding to the reaction mixture obtained by reacting **1**
with **2a** and **4c** in the presence of reducing agent and **7a**, see
Fig. 4b. In each step, the RSI assessed the pathways correctly and
the stopper pathways were always scored with the smallest RSIs
helping validate the effectiveness of the approach. As the number
of generations increases, the range of the RSIs decreases due to
the more diluted reaction mixtures analysed. However, in this

case, this factor did not limit the selection of the most reactive
pathway. During the three-step experiment run, the in-line ESI-
MS measurements were automated in order to validate the
detection of the expected third-generation products in the final
four reaction mixtures screened in real time. However, the ESI-
MS data of the final four reaction mixtures were inconclusive,
highlighting the different diagnostic sensitivity and consequently
the limitations in the analysis of complex reaction mixtures (like
those studied and presented in this work) of this technique when
compared to the infrared spectroscopy. To confirm this, and to
also validate those pathways independently from the system, we
conducted ESI-MS measurements of the reaction mixtures
starting from the second-generation product **6c**. The measure-
ments conclusively show that the presence of the third-generation
compounds could be detected by ESI-MS, after purification by
column chromatography. It was therefore possible to confirm also
for this set of reactions, the reliable use of RSI to implement in
similar decision-making platforms.

**Conclusions**. In summary, we have found that it is possible to
correlate the RSI with the level of reactivity of the chemical
transformations and to use this value for the navigation of
chemical spaces without necessarily having a target-oriented
strategy *a priori*. We first demonstrated the ability to navigate a
two-step reaction network with high atom economy, using the
RSI to assess changes between the starting materials and the
products of reactions as proof of principle. Building on this, it was
possible to demonstrate that the RSI can be directly related to the
reactivity of a chemical transformation. In addition, it is also
possible to observe a significant difference, equal to an order
of magnitude, between reactive and non-reactive pathways when
the complexity of the reaction mixtures is contained. We then

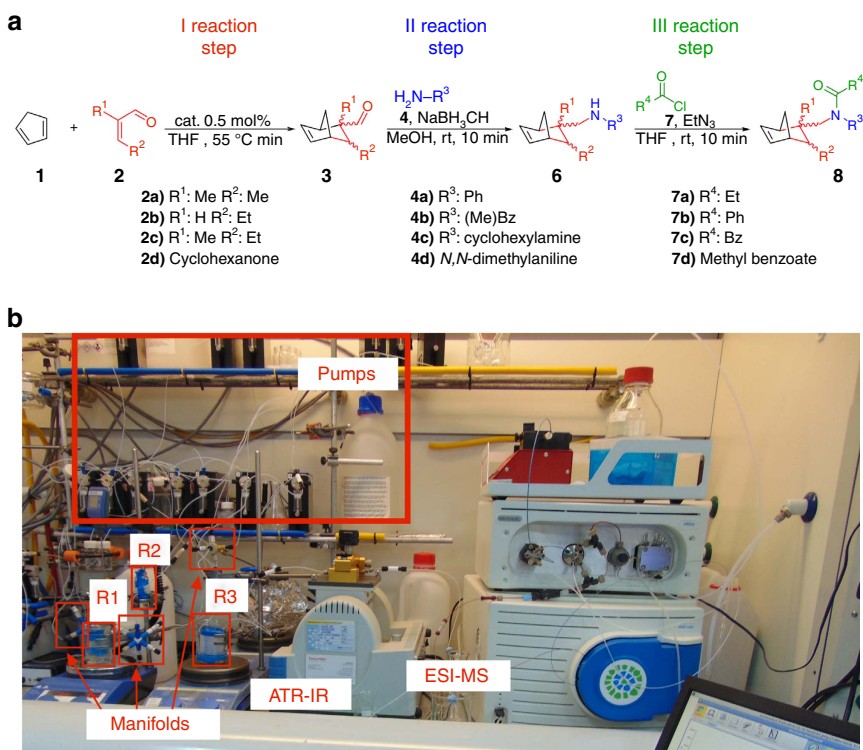

**Figure 3 | Reaction steps and physical setup of the pathway-dependent chemistry platform.** (**a**) Schematic of the three reaction steps selected to build this chemical network. The exploration of the reaction network is obtained by reacting a core molecule (**1**) in three-step reactions with four different reagents in each step (compounds **2a–d**, **4a–d** and **7a–d**, respectively). (**b**) Photograph of the pathway-dependent chemistry platform. The setup of this platform consists of 14 computer-controlled syringe pumps, three flow reactors (connected with each other using polytetrafluoroethylene tubing and manifolds), a bench-top infrared spectrometer equipped with an ATR-IR flow cell and a bench-top ESI-MS.

developed a decision-making platform, where the RSI is used to navigate a $4 \times 4 \times 4$ reaction network involving also the use of two different solvents. With this approach, we were able to demonstrate that it is possible to efficiently reduce the sampling to a smaller space by focussing on the level of reactivity of the chemical transformations. After the full run of the first reaction step, the duration of the screening and material waste can be significantly reduced in the second step and even more in the third step. Indeed, with this methodology, it was possible to reduce the screening time from 12 to 5 h *ca.* for the second-generation screening process and from 44 to 8 hours *ca.* for the third generation. Furthermore, a theoretical analysis of this approach shows that the time to navigate a given network dramatically decreases as the fraction of the total number of reactions that need to be performed when the number of reagents per generation ($R$) and reaction generations ($G$) increase. In fact, with this approach, the number of reactions required only increases as a fraction of the total number of possible reactions, and this increases as a function of the number generations and hence decision points. For example, here only 16 out of a total 84 reactions needed to be explored, but this goes down much further as the number of generations, and hence decision points, increase (see Supplementary Fig. 5).

We also demonstrated that the analysis of complex mixtures with the RSI reliably leads to the selection of the most reactive pathway. This is of particular interest as the reactivity can be assessed even from the very small differences in similar infrared spectra. The possibility of analysing reaction mixtures in real time

and without any work-up or purification steps makes this methodology highly attractive, when an autonomous and fast screening of many chemical combinations under different conditions is required. The unique association of the RSI with the reactivity of chemical transformations makes this platform a suitable choice for the navigation of the chemical space focussing on the region leading to the highest reactivity of a reaction network under study. This methodology could be further expanded to become a discovery tool, but in future, we will need to evaluate this approach next to more traditional approaches to see whether discoveries can be made more quickly and be more 'exciting' or unpredictable if reactivity is followed rather than explicit target-based searching. We think that this approach has promise due to the possibility of using the RSI to discriminate between reactive and non-reactive substrates for one-step reactions, as this will afford the possibility of using the RSI to navigate reaction networks without the generational constrains. In this way, the RSI could be used to help drive dedicated chemical discovery systems that follow reactivity rather than the normal rules of synthetic chemistry and retrosynthesis, leading to new approaches to chemical robotics[36,37].

## Methods

**Materials and chemicals.** Solvents for synthesis (AR grade) were supplied by Fisher Chemicals and Riedel-de Haen. Deuterated solvents were obtained from Goss Scientific Instruments Ltd. and Cambridge Isotope Laboratories Inc. All other reagents were supplied by Sigma-Aldrich Chemical Company Ltd., Fisher Chemicals and Lancaster Chemicals Ltd. All commercial starting materials were used as supplied, without further purification, with the only exception

being dicyclopentadiene, which was freshly cracked in order to form 1,3-cyclopentadiene[38]. Polytetrafluoroethylene tubing with different internal diameters, PEEK connectors and manifolds were supplied by Kinesis Ltd.

**Table 1 | RSI and conversion values of a Diels–Alder reaction conducted under flow conditions using different temperatures and residence time.**

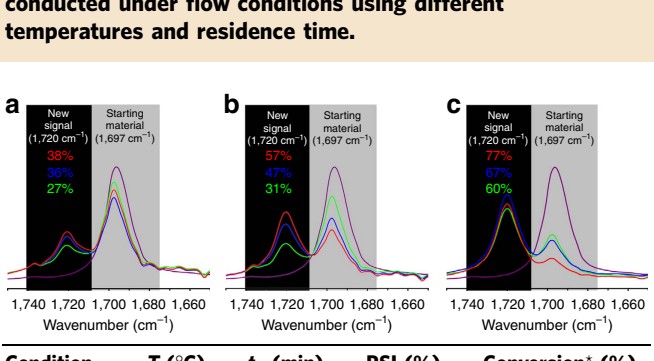

| Condition | $T$ (°C) | $t_R$ (min) | RSI (%) | Conversion* (%) |
|-----------|----------|-------------|---------|-----------------|
| I   | 20 | 7  | 3.7  | 27 |
| II  |    | 13 | 5.2  | 36 |
| III |    | 20 | 5.5  | 38 |
| IV  | 35 | 7  | 4.4  | 31 |
| V   |    | 13 | 11.6 | 47 |
| VI  |    | 20 | 15.0 | 57 |
| VII | 55 | 7  | 16.0 | 60 |
| VIII|    | 13 | 18.1 | 67 |
| IX  |    | 20 | 20.5 | 77 |

Three graphs illustrating the infrared spectra of the experiments at 20 °C (**a**), 35 °C (**b**) and 55 °C (**c**). In each graph, the infrared spectra of **3a** obtained at 7, 13 and 21 min are shown (green, blue and red spectra, respectively) and are compared with the starting material spectra (in purple).
*Conversions calculated according to curve-fitting operations of the aldehyde peaks in the ATR-IR spectra of reaction mixtures and confirmed by [1]H-NMR spectroscopy.

**NMR spectroscopy.** All NMR data were recorded on a Bruker Advance 500 MHz. [1]H-NMR at 500 MHz and [13]C-NMR at 125 MHz, in deuterated solvent, at $T = 298$ K, using tetramethylsilane as the scale reference. Chemical shifts are reported using the $\delta$-scale, referenced to the residual solvent protons in the deuterated solvent for [1]H- and [13]C-NMR (that is, [1]H: $\delta$ (CDCl$_3$) = 7.26; [13]C: $\delta$ (CDCl$_3$) = 77.16). All chemical shifts are given in p.p.m. and all coupling constants ($J$) are given in Hz ($J$) as absolute values. Characterization of spin multiplicities: s = singlet, d = doublet, t = triplet, q = quartet, m = multiplet, dd = double doublet, dt = double triplet, dq = double quartet, and ddt = double doublet of triplets.

**Gas chromatography mass spectrometric measurements.** Gas chromatography mass spectrometric analysis was performed using an Agilent Technologies 7890A GC system equipped with Agilent Technologies 5975C inert XL MSD with Triple-Axis Detector. The column used was Agilent 19091N-102: 260 °C, 25 mm × 200 μm × 0.2 μm wide bored.

**Syringe pumps.** The control over the fluids was performed using the C3000 model and TriContinent pumps (Tricontinent Ltd, CA, USA) equipped with 5 ml syringes (TriContinent) according to the requirements of the experiments.

**In-line ATR-IR spectroscopy.** All spectra were recorded on a Thermo Scientific Nicolet iS5 Fourier transform infrared equipped with a ZnSe Golden Gate ATR-IR flow cell. Frequencies are given in cm$^{-1}$. The resolution was set at 4 cm$^{-1}$ and each sample spectrum was recorded with 21 scans.

**Bench-top mass spectrometry.** The spectra were recorded using a Microsaic systems 4000 MiD spraychip (electrospray ionization source). Masscape software was used for control of sample methods and data analysis. The specifications of this spectrometer are listed in Supplementary Table 1.

**Flow setup and algorithm.** A custom-made developed LabVIEW application (provided by National Instruments Corp.) was employed to program the pumps in order to deliver the desired flow-rates and to control the infrared spectroscopy. A summary of the setup characteristic and the setup schematic are illustrated in Supplementary Table 2 and Supplementary Fig. 1, respectively. All solutions were pumped by TriContinent pumps equipped with 5 ml syringes. All syringe pumps

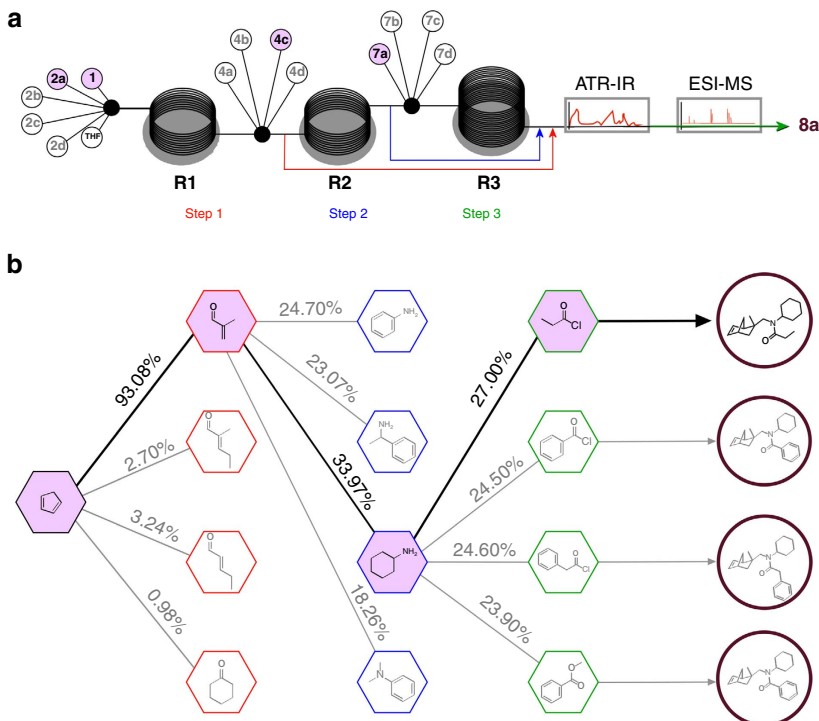

**Figure 4 | Results/summary of the navigation of the model reaction network using the RSI algorithm.** (**a**) Schematic of the organic reaction search engine used for the navigation of the 4 × 4 × 4 chemical network. Each reactant is associated with a syringe pump and connected to the flow reactor of the first, second or third step, using a multi-way connector. After each reaction step, the RSI for the pathway screened is calculated and used to continue the reaction network navigation. (**b**) Illustration of the reaction network navigation using the RSI showing the navigation of the III generation compounds **8a–d** synthesis and the selection of the pathway related to compound **8a** using the RSI.

were associated with one reagent and were equipped with polytetrafluoroethylene tubing for delivering the solutions into the flow reactors. The syringe pumps of the first-generation reactants, the core molecule and the solvent are connected to the first flow reactor (**R1**) using a eight-way connector. **R1** is then connected to the second flow reactor (**R2**) together with the four syringe pumps associated with the second-generation reagents, using a six-way connector. **R2** is also connected with a six-way connector to the third flow reactor (**R3**) together with the four syringe pumps associated with the third-generation reactants. **R3** is then connected to the portable ATR-IR with a two-way connector. The ATR-IR spectrometer is then connected to the portable ESI-MS using another two-way connector. The physical connection between the pumps and the computer was built based on wiring RS232 cables to a multi-block assembly. In-line analytics were physically connected to a computer by a USB to Serial connection. In particular, the communication between the computer and the ATR-IR spectrometer was enhanced by controlling the property software OMNIC via ActiveX Functions, available as LabVIEW features. The ESI-MS was automated using the property software Microsaic Masscape (Version 2.52). The language used to communicate between the PC and the pumps was originally created by the pump manufacturer. A schematic of the flow diagram of this algorithm is illustrated in Supplementary Fig. 2. Details on the algorithm—including data processing operations—are reported in Supplementary Note 1, Supplementary Equations 1–5, Supplementary Fig. 3 and Supplementary Tables 3–16.

**Multi-step reactions: navigation up to the second generation.** The compounds and their preparation used are described in Supplementary Table 17. In each of the first group of three reactions, 1.5 ml of the reagent associated with P1 at a flow rate of $0.075\,\mathrm{ml\,min}^{-1}$ and combined with 1.5 ml of one of the reagent associated with P2, P3, P4 or P5 (randomly selected) at the same flow rate and then heated to 55 °C in reactor **R1** + **R2** ($t_R = 20\,\mathrm{min}$). Between one reaction and the next, 1.5 ml of solvent from P14 was passed through **R1** + **R2** at a flow rate of $0.15\,\mathrm{ml\,min}^{-1}$, resulting in a waiting time of 10 min, during which the flow stream passed through an ATR-IR flow cell in order to collect its infrared spectrum and calculate its MSE value. At the end of the third reaction and washing cycle, the system calculates the RSI values of all experiments of the same generation and compares them, in order to select the biggest one and the pathway, as pump combination, associated with it—which in this case was the product obtained from mixing the reagents with P1 and P2, corresponding to compound **3a**. In each of the second group of three reactions, 1.125 ml of the reagent associated with P1 at a flow rate of $0.0325\,\mathrm{ml\,min}^{-1}$ and combined with 1.125 ml of the reagent associated with P2 at the same flow rate and then heated to 55 °C in reactor **R1** ($t_R = 20\,\mathrm{min}$). The output from **R1** was combined with 0.75 ml of one of the reagent associated with P6, P7, P8 or P9 (randomly selected) at the flow rate of $0.75\,\mathrm{ml\,min}^{-1}$ in **R2** ($t_R = 10\,\mathrm{min}$). Between one reaction and the next, 1.5 ml of solvent from P14 was passed through **R1** + **R2** at a flow rate of $0.15\,\mathrm{ml\,min}^{-1}$, resulting in a waiting time of 10 min, during which the flow stream passed through an ATR-IR flow cell in order to collect its infrared spectrum and calculate its MSE value. At end of the third reaction and washing cycle, the system calculates the RSI values of all experiments of the same generation and compares them, in order to select the biggest one and the pathway, as pump combination, associated with it—that in this case was the product obtained from mixing the reagents in P1, P2 and P8, corresponding to compound **5c**. The eight reaction mixtures collected were further analysed by NMR spectroscopy and MS (see Supplementary Note 4). The operations executed with VI-4 for each sequence run are summarized in Supplementary Note 2 and a list of the experiments ran with conversion, RSI and MSE value for each of the six reactions is reported in Supplementary Table 18.

**Multi-step reactions: navigation up to the third generation.** The compounds and their preparation used are described in Supplementary Table 19. In each of the first group of three reactions, 3 ml of the reagent associated with P1 at a flow rate of $0.15\,\mathrm{ml\,min}^{-1}$ and combined with 3 ml of one of the reagent associated with P2, P3, P4 or P5 (randomly selected) at the same flow rate and then heated to 55 °C in reactor **R1** + **R2** + **R3** ($t_R = 20\,\mathrm{min}$). Between one reaction and the next, 1.5 ml of the solvent from P14 was passed through **R1** + **R2** + **R3** at a flow rate of $0.3\,\mathrm{ml\,min}^{-1}$, resulting in a waiting time of 10 min, during when the flow stream passed through an ATR-IR flow cell in order to collect its infrared spectrum and calculate its MSE value. At the end of the third reaction and washing cycle, the system calculates the RSI values of all experiments of the same generation and compares them, in order to select the biggest one and lock the pump combination associated with it—which in this case was the product obtained from mixing the reagents with P1 and P2, corresponding to compound **3a**. In each of the second group of three reactions, 1.125 ml of the reagent associated with P1 at a flow rate of $0.0325\,\mathrm{ml\,min}^{-1}$ and combined with 1.125 ml of the reagent associated with P2 at the same flow rate and then heated to 55 °C in reactor **R1** ($t_R = 20\,\mathrm{min}$). The output from **R1** was combined with 0.75 ml of one of the reagent associated with P6, P7, P8 or P9 (randomly selected) at the flow rate of $0.75\,\mathrm{ml\,min}^{-1}$ in **R2** ($t_R = 10\,\mathrm{min}$). Between one reaction and the next, 4.5 ml of solvent from P14 was passed through **R1** + **R2** + **R3** at a flow rate of $0.15\,\mathrm{ml\,min}^{-1}$, resulting in a waiting time of 30 min, during which (after the first 10 min) the flow stream passed through an ATR-IR flow cell in order to collect its infrared spectrum and calculate its MSE value. At the end of the third reaction and washing cycle, the system

calculates the RSI values of all experiments of the same generation and compares them, in order to select the biggest one and the pathway, as pump combination, associated with it—that in this case was the product obtained from mixing the reagents in P1, P2 and P8, corresponding to compound **6c**. In each of the third group of three reactions, 1.5 ml of the reagent associated with P1 at a flow rate of $0.0325\,\mathrm{ml\,min}^{-1}$ and combined with 1.5 ml of the reagent associated with P2 at the same flow rate and then heated to 55 °C in reactor **R1** ($t_R = 20\,\mathrm{min}$). The output from **R1** was combined with 1.125 ml of the reagent associated with P5 at the flow rate of $0.75\,\mathrm{ml\,min}^{-1}$ in **R2** ($t_R = 10\,\mathrm{min}$). The output from **R2** was combined with 0.75 ml of one of the reagent associated with P10, P11, P12 or P13 (randomly selected) delivered at the flow rate of $0.15\,\mathrm{ml\,min}^{-1}$ in **R3** ($t_R = 10\,\mathrm{min}$). Between one reaction and the next, 1.5 ml of solvent from P14 was passed through **R1** + **R2** + **R3** at a flow rate of $0.3\,\mathrm{ml\,min}^{-1}$, resulting in a waiting time of 5 min, which allows the stream to pass through an ATR-IR flow cell in order to collect its infrared spectrum and calculate its MSE value. At the end of the third reaction and washing cycle, the system calculates the RSI values of all experiments of the same generation and compares them, in order to select the biggest one and the pathway, as pump combination, associated with it—which in this case was the product obtained from mixing the reagents in P1, P2, P5 and P10, corresponding to compound **8a**. The 12 reaction mixtures collected were additionally analysed by NMR spectroscopy and MS where possible (see section 4.4 for the characterization of these nine compounds). The operations executed with VI-4 for each sequence run are summarized in Supplementary Note 3, a list of the experiments ran with conversion, RSI and MSE value for each of the nine reactions is reported in Supplementary Table 20 and the comparison between the sum of the starting materials and the experimental infrared is summarized in Supplementary Fig. 4.

**Syntheses of the compounds isolated from the reaction network navigated.** Most of the products were not isolated due to the approach that includes a system without any purification. Therefore, in most of the cases, the analysis was performed directly on the crude reaction mixtures to ensure the reaction was proceeding as planned. Analysis of the crude reaction mixtures did not allow the unambiguous assignment of $^1$H- and $^{13}$C-NMR signals, and all signals assigned to the reaction products are given here. For more details, see Supplementary Note 4.

**Data availability.** The data that support the findings of this study are available from the corresponding author upon reasonable request.

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

## Acknowledgements

We gratefully acknowledge financial support from the EPSRC (Grant Nos. EP/H024107/1, EP/I033459/1, EP/J00135X/1, EP/J015156/1, EP/K021966/1, EP/K023004/1, EP/K038885/1, EP/L015668/1, EP/L023652/1), BBSRC (Grant No. BB/M011267/1), ERC (project 670467 SMART-POM) and the Royal-Society Wolfson Foundation for a Merit Award to L.C. and the University of Glasgow. J.M.G. acknowledges financial support from the Polish Ministry of Science and Higher Education Grant No. 1295/MOB/IV/2015/0.

## Author contributions

L.C. conceived the idea, designed the project and coordinated the efforts of the research team. V.D. built the system with the initial help from V.S. in designing and programming the flow system and conducted the experiments with input from A.B.H. and J.M.G. V.D. and L.C. co-wrote the paper with input from all the authors.

## Additional information

**Competing interests:** L.C. is a director of, and owns some shares in, Cronin Group PLC, set up to commercialize new approaches to design, discovery and digitization in chemistry. The remaining authors declare no competing financial interests.

