## [Peer review file · Nature Communications]

Reviewers' comments:

Reviewer #1 (Remarks to the Author):

Overall recommendation.

This manuscript presents an impressive use of flow chemistry, inline monitoring, and real-time feedback to autonomously explore the reactivity of set of molecular building blocks. Reactivity, as judged by a difference in the IR spectra of the products and reactants, is used as a measure of the success of a reaction, as opposed to the more traditional measures such as yield of a target molecule. This measure allows the autonomous ranking of trial reactions and is used to guide the selection of reaction pathways to form more complex products. To my knowledge, this is the first example of an autonomous system using an algorithm to discover new organic molecules, as opposed to optimising synthesis, without user input beyond providing the starting materials.

The advantage of this method is that detailed product analysis is not necessary, enabling automatic decision making; knowledge of the chemistry itself is not required; and many fewer reactions are necessary to discard 'dead end' possibilities in a reaction set.

The use of reactivity and 'reaction screening index' (RSI) as a proxy for reaction success is justified by the authors as a rapid screening method, although discussion of the possible drawbacks is absent. The chief issue to me is that reactivity does not necessarily mean the desired outcome has occurred; indeed, many of the reactions described have significant quantities of side products present – e.g. see pages S34-35. This is not discussed in detail in the manuscript – I would have liked the question of whether reactivity is always a valid method of screening for reaction success to be more fully explored. One possible other weakness of the study is that the molecules formed do not have a function or targeted goal (or this is not discussed); the reactions chosen are well known and studied, and there isn't a discussion of the general applicability of the approach. The addition of a paragraph describing the broader scope of the methods outlined, and more detail of the advantages this method has over current methods of discovery, would help to alleviate this criticism.

The system can correctly and autonomously identify non-reactive 'stoppers' that are used as control experiments in this context, but with less of a margin as the number of generations screened increases (i.e. for generation 1, the RSI for all four options screened ranges from 0.98 % - 93.08%; for generation 2, the RSI for all four options screened ranges from 18.26 % - 33.97%, and for generation 3, the RSI ranges from 23.90% - 27.00%, Table S19). This may be an inevitable consequence of the way the process runs (i.e. each step adds more 'difference' from the spectrum of the original starting materials), but this is not discussed in detail. Likewise, the rationale for discarding pathways with a small difference in RSI to the most reactive step is not discussed and may not be justified, despite the resultant minimising of the number of reactions to be performed.

On balance, this work deserves publication in Nature Communications, with more discussion on the areas mentioned above, and would be of interest to people in flow chemistry, automation, online analysis, and organic synthesis. Inline analysis used has been previously reported (see, for example, references in the review by Sans & Cronin, Chem Soc Rev, 2016, 45, 2032-2043). The use of algorithms for automated optimisation of synthesis has been previously reported, by the Ley (Org Proc Res Dev, 2016, 20, 386-394), Cronin (Chem Sci, 2015, 6, 1258) and Bourne (React Chem & Eng, 2016, 1, 366-

371) groups, among others (see examples in Curr Opp Chem Eng, 2015, 9, 1-7). Algorithmic automation of discovery of new chemistries has been reported by the Cronin group in inorganic chemistry (Nat Comm, 5, 3715-3723), and, to my eye, similar methods have been used here for organic synthesis, albeit with a different success criteria. Hence, the main novelty of the paper is the use of 'reactivity' as a success measure, although this may be controversial for organic chemists who are accustomed to unwanted reactivity leading to products that are undesired, unhelpfully diverse, or vastly outnumbered by impurities! Despite this, researchers in the field could use this work to inform new ways of automating synthesis, with the ultimate goal of exploring a wide chemical parameter space with minimal user input. I am convinced that automation and algorithms will become more widely used in chemical synthesis, and this work is an important forerunner.

A few minor additional points:

Reference 16 and 21 in the manuscript are the same.

Reproduction of more than one of the IR spectra described in the manuscript would be very beneficial to give an idea of the differences picked up by the algorithm in cases where the difference is small (i.e. compared across generations or those for which RSI values have been quoted). I would have liked to see more discussion of the sensitivity of this approach. Would this method be applicable to all chemistries or just the subset produced here? Are most products sufficiently different from their starting materials to be picked up by this technique?

Reproduction of the NMR spectra of the molecules synthesised would also be desirable to illustrate relative percentages of minor and major products.

A diagram of the flow pathway, although complex, would aid understanding more than a photograph.

I was not convinced by the mathematical discussion of the number of pathways. Clearly, if you increase the number of generations, only selecting one possible pathway, you dramatically decrease the total fraction of experiments you have to do. However, this doesn't address the possible drop in difference between the RSIs seen by generation 3, which, if I've understood correctly, may be a limiting factor in the number of generations/reaction steps you can perform. This also doesn't address whether selecting only one pathway is always the best option, or whether discarding potentially interesting alternatives might occur with this approach.

Reviewer #2 (Remarks to the Author):

The paper describes a technically interesting system, but to me does not bring out the point of the work very well. (I enjoyed reading the supplementary more than the paper).

The paper seems to be presented as a way to make novel compounds using several sequential reactions without intermediate work-up / purification (or at least to identify which of many possible products is from the most 'reactive' pathway).

The paper uses the odd term 'Reactive' as it is essentially following loss of starting material, not formation of product (or products). For multi-step synthesis the interest is

much more in selectivity (and consequently, yield).

It might perhaps be more clearly presented as a way to discover new reactivity (as it just looks a loss of starting materials) in which case the need for several steps needs to be justified. Perhaps the case can be made that the penultimate steps provide a more complex (and not commercially available) precursors for the last stage?

The approach would be more convincing in a scenario where most of the reactions do not work.

The abstract is poor - more on an introduction. It needs to concisely state what the research has delivered.

"However, the autonomous exploration of chemical space has been limited by the need for separate work-up-separation steps searching for molecules rather than reactivity." Which seem to be saying that looking for what is formed is not as good as looking at what has reacted - something I suspect most chemists would disagree with. The 'work-up-separation' is irrelevant - crude products can be analysed in many ways for selectivity in product formation.

"target-driven molecular synthesis, searching chemical databases" What is the relevance of the last part - how does it link to the first?

"Herein we present a reactivity explorer robot that is capable of navigating a large reaction network and assessing the reactivity of the chemical transformations autonomously, without needing to do every reaction." And later "autonomously assess, and rank the reactivity of all reagent combinations"

It does not assess the reactivity of all the transformations - it just picks the best from the first step, and only tries the second step reagents on this one (so gets no information on the other possibilities). There is no reason why reactivity in the second step should be correlated to that in the first step (except when the first step has failed in a way that the functionality expected for the second step is not present). Better to say "attempts to find an overall sequence of reactions where every stage results in maximum loss of starting material (with the reasonable hope that it also maximize production of a particular product).

What does Figure 2 add - to me is just confuses as it seems to imply that many reaction paths have been examined). Fig 1 at least shows the pruning being done. The comparison to the combinatorial approach in Fig 1 might be more useful if it showed the reality - in such a combiChem approach many of the routes would fail (possibly in the first step) so only a proportion (probably small) of the third generation compounds would be as required. Method B might only make 4 compounds, but at least the chance of these 4 being correct is higher as the 'best' generation 1 and 2 reactions have been selected.

Technically the authors look at the least squares difference between reaction product

and the spectra which would be expected from starting materials (including the crude products from previous steps).

This only approximates to the loss of starting material as the extent to which the products change the region of the IR spectra observed matters as well (i.e. different products will have different effects so the RMS difference for the same extent of reaction will be different in different cases). For example I suspect that all three acid chlorides in the last step reacted to completion - so the difference in RSI is due to the differing products not differing reactivity's. Using GC or UPLC to monitor disappearance of starting material would be more reliable and since only a single data point is taken, not significantly more challenging. .

The 'validation' (Table 1) uses the very special case where a strongly IR absorbing group (a carbonyl) shifts between starting material and product. It also uses the same reaction under different conditions so the same products are presumably being formed, just to a different extent- quite different to the main topic of the paper where different products are being formed.

I do not object to the technique, but the problems need to be acknowledged.

Needs to be careful with the spelling of chemical names:

methacroleine (2a), trans-2-methyl-pent-2-ene

References 15 and 20 are the same as are 16 and 21!

Overall I feel that the work is well worth publishing, but with most of the hype taken out and consequently a much clearer description of what has been achieved.

Reviewer #3 (Remarks to the Author):

Review of NCOMMS-17-00922

This is a very interesting and novel paper. I think that it is definitely publishable and potentially suitable for Nature Communications. The concept of using automated reactors in this way is very promising and clearly has wider applications beyond this demonstration of concept. I have two serious points that need to be addressed.

1. There is no description or diagram of the equipment which would enable anyone else to replicate the work. Figure S1 is a not particularly good photo which does little more than give an impression of the equipment. What is needed is a clear piping diagram with a list of components/suppliers, showing how everything is interconnected so that others could repeat the work if they wished.

2. I feel that there is too much hype in the title "An autonomous organic reaction robot that searches for chemical reactivity". Clearly, they are using quite a complex set of pumps but I don't think that one is justified in describing such a set up as a "robot". Similarly the algorithms, although clever, are not really artificial intelligence in the sense that I understand. Also, the statement on page 2, line 8/9 about "the discovery of new reactions and molecules is still firmly in the domain of the chemist" is perhaps over dogmatic. For example, the paper "Automated Serendipity with Self-Optimizing

Continuous-Flow Reactors" by Amara et al. EUROPEAN JOURNAL OF ORGANIC CHEMISTRY, 2015, 6141, DOI: 10.1002/ejoc.201500980, describes the discovery of a previously unknown reaction.

In summary, this Communication presents an excellent piece of work that needs to be described more fully in the Supplementary Information and is good enough to stand by itself without the inflated claims in the title and some of the terminology.

(A minor point: The use of the abbreviations ESI, RSI and SI all in the same paper is slightly confusing).

Replies to referee comments. Referee comments in italics, our replies in normal type. Additions to the MS highlighted in yellow.

Reviewer #1 (Remarks to the Author):

Overall recommendation.

This manuscript presents an impressive use of flow chemistry, inline monitoring, and real-time feedback to autonomously explore the reactivity of set of molecular building blocks. Reactivity, as judged by a difference in the IR spectra of the products and reactants, is used as a measure of the success of a reaction, as opposed to the more traditional measures such as yield of a target molecule. This measure allows the autonomous ranking of trial reactions and is used to guide the selection of reaction pathways to form more complex products. To my knowledge, this is the first example of an autonomous system using an algorithm to discover new organic molecules, as opposed to optimising synthesis, without user input beyond providing the starting materials. The advantage of this method is that detailed product analysis is not necessary, enabling automatic decision making; knowledge of the chemistry itself is not required; and many fewer reactions are necessary to discard 'dead end' possibilities in a reaction set.

The use of reactivity and 'reaction screening index' (RSI) as a proxy for reaction success is justified by the authors as a rapid screening method, although discussion of the possible drawbacks is absent. The chief issue to me is that reactivity does not necessarily mean the desired outcome has occurred; indeed, many of the reactions described have significant quantities of side products present – e.g. see pages S34-35. This is not discussed in detail in the manuscript – I would have liked the question of whether reactivity is always a valid method of screening for reaction success to be more fully explored.

We do agree with the reviewer “*that reactivity does not necessarily mean the desired outcome has occurred*”. However, in this work we present a non-targeted oriented methodology, which focuses on the searching chemical space for maximum reactivity. So if the system finds maximum reactivity then it has done its job. The utility of this approach of course needs to be evaluated and we are following up on this. We have added the following text in the conclusions:

This methodology could be further expanded to become a discovery tool but in future we will need to evaluate this approach next to more traditional approaches to see if discoveries can be made more quickly, and be more 'exciting' or unpredictable if reactivity is followed rather than explicit target based searching.

One possible other weakness of the study is that the molecules formed do not have a function or targeted goal (or this is not discussed); the reactions chosen are well known and studied, and there isn't a discussion of the general applicability of the approach.

The focus of our study is not placed on the chemistry but instead on the new chemical space exploration methodology that aims to focus the search on the reactive islands of the chemical space. The chemistry has been targeted for the different range of reactivity and mixture complexity to analyse. In addition, our model chemical system includes three fundamental and commonly used reactions for the synthesis of many organic molecules (including drugs) to showcase the broad applicability of this methodology in organic and medicinal chemistry. The chemistry has been selected because it is very interesting for organic synthesis and yet is well-known, what facilitates the interpretation of the results. The methodology has a virtually unlimited scope in synthetic space and a number of complementary in-line analytical techniques can be employed to assess the reactivity. Indeed, we have developed some in-line analytical techniques (e.g. Chem. Sci. 2015, 6, 1258) which can complement the current study and open new frontiers in the search of new reactivities.

The addition of a paragraph describing the broader scope of the methods outlined, and more detail of the advantages this method has over current methods of discovery, would help to alleviate this criticism.

We thank the referee for this comment. It has been now made more explicit that we present a new methodology that works in an uneven space. In fact, we cut the chemical space after each step to finally focus on its overall most reactive region. Each step-selected contribute to the final selection as we are going step by step closer to the identification of the most reactive pathway in that highly reactive region of the space. The following text has been added:

To date, the development of automated systems in chemistry³⁵ has focused on the generation of pre-designed libraries, whilst the real time experimental exploration of chemical reactivity remains underexplored. This is because the process of organic synthesis is normally target-based, and reactivity searches are normally focused on a particular transformation / optimization of a given reaction step.²⁴²⁷ We therefore hypothesized that the 'closed-loop' exploration of chemical space could be implemented using a simple approach using spectroscopic feedback focusing on the differences between the starting reagents and the products enabling the search for new molecules, reactions and synthetic pathways. This is because the ability to search for reactivity, rather than following the constraints of a design-to-target approach might lead to the discovery of new reactions and molecules by following reactivity first. This approach could allow the elimination of bias which can prevent the human experimenter from doing a particular set of experiments. By focusing on a new metric based approach following chemical change, the system will be able to explore without bias thereby allowing machine learning based upon only sensor feedback rather than relying on databases. Ideally a system that could allow both searching of new chemical space, and the ability to update the database to predict new routes would be the most powerful combination.³⁶

The system can correctly and autonomously identify non-reactive 'stoppers' that are used as control experiments in this context, but with less of a margin as the number of generations screened increases (i.e. for generation 1, the RSI for all four options screened ranges from 0.98 % - 93.08%; for generation 2, the RSI for all four options screened ranges from 18.26 % - 33.97%, and for generation 3, the RSI ranges from 23.90% - 27.00%, Table S19). This may be an inevitable consequence of the way the process runs (i.e. each step adds more 'difference' from the spectrum of the original starting materials), but this is not discussed in detail.

This is a very good point and occurs because the sample gets diluted with subsequent generations. It could be possible to add in-line purification methods to concentrate the samples and perhaps eliminate undesired materials (e.g. salts in an organic reaction). This would enhance the signal and therefore lead to higher sensitivity allowing for a larger number of steps to occur; however we will explore this in later work. To draw attention to this potential issue we have added the following comment:

It is worth noting that as the number of generations increase, the range of the RSIs decreases due to the increase in dilution however in this did not limit the selection of the most reactive pathway.

Likewise, the rationale for discarding pathways with a small difference in RSI to the most reactive step is not discussed and may not be justified, despite the resultant minimising of the number of reactions to be performed.

We thank the referee for this comment. It has been now made more explicit that this is due to a choice of methodology (and can be expanded later). Here we are looking for the highest reactive pathways rather than for the one forming a specific outcome in highest yields. We demonstrate that this methodology is also able to give a good approximation of which pathway will yield the highest reactivity of the starting materials. Also by keeping the algorithm simple we keep the system autonomous which is important in ensuring that additional complex work-up steps, needing human intervention, are minimised. To explain this we added the following comment:

The idea is not just to follow yield, but to see how following maximum reactivity can lead to linking together many reactions with minimum work-up allowing the system to autonomously follow reactivity.

On balance, this work deserves publication in Nature Communications, with more discussion on the areas mentioned above, and would be of interest to people in flow chemistry, automation, online analysis, and organic synthesis.

Inline analysis used has been previously reported (see, for example, references in the review by Sans & Cronin, Chem Soc Rev, 2016, 45, 2032-2043). The use of algorithms for automated optimisation of synthesis has been previously reported, by the Ley (Org Proc Res Dev, 2016, 20, 386-394), Cronin (Chem Sci, 2015, 6, 1258) and Bourne (React Chem & Eng, 2016, 1, 366-371) groups, among others (see examples in Curr Opin Chem Eng, 2015, 9, 1-7). Algorithmic automation of discovery of new chemistries has been reported by the Cronin group in inorganic chemistry (Nat Comm, 5, 3715-3723), and, to my eye, similar methods have been used here for organic synthesis, albeit with a different success criteria.

We thank the reviewer for these suggestions; we have now included these references in the manuscript - Refs 8, 21, 22, 27, 28, 34

Hence, the main novelty of the paper is the use of 'reactivity' as a success measure, although this may be controversial for organic chemists who are accustomed to unwanted reactivity leading to products that are undesired, unhelpfully diverse, or vastly outnumbered by impurities! Despite this, researchers in the field could use this work to inform new ways of automating synthesis, with the ultimate goal of exploring a wide chemical parameter space with minimal user input. I am convinced that automation and algorithms will become more widely used in chemical synthesis, and this work is an important forerunner.

A few minor additional points:

Reference 16 and 21 in the manuscript are the same.

We apologize for this repetition; this has now been corrected.

Reproduction of more than one of the IR spectra described in the manuscript would be very beneficial to give an idea of the differences picked up by the algorithm in cases where the difference is small (i.e. compared across generations or those for which RSI values have been quoted).

We agree with the referee and a new figure, illustrating the difference between the experimental and starting materials IR spectra of the three generations reaction has been added in the supporting information (Figure S3).

I would have liked to see more discussion of the sensitivity of this approach. Would this method be applicable to all chemistries or just the subset produced here? Are most products sufficiently different from their starting materials to be picked up by this technique?

We think that IR spectroscopy is a suitable technique to look at the reactivity of organic molecules. In fact, contrary to the more diagnostic NMR or MS techniques (primary choice for the identification of reaction outcomes), IR spectroscopy permits to better map the changes of the starting materials and hence to correlate this parameter to their reactivity, rather than to the reaction yields. Furthermore, the employment of IR allows the detection of a large number of vibrational modes, and therefore is applicable to a much broader range of chemistries than NMR (limited to a set of nuclei) or MS (only charged species). Answering the second question, the products are sufficiently different to the starting materials as evidenced by the RSI values obtained. The reviewer raises a very important point. If the RSI indexes are not different enough to make a decision, then other in-line techniques (UV-Vis, NMR, MS, Raman, etc.) could be employed. All those techniques can be easily added together to yield complementary information about the reactivity of the system via a rolling PCA or similar.

Reproduction of the NMR spectra of the molecules synthesised would also be desirable to illustrate relative percentages of minor and major products.

The ^1H NMR used for the calculation of the conversion yields of the reaction mixtures corresponding to the formation of compounds **3a-d** and **5a-d** have now been included in the supporting information (section 4.5).

A diagram of the flow pathway, although complex, would aid understanding more than a photograph.

We clarified the schematic representation of the flow setup in Figure 4A.

I was not convinced by the mathematical discussion of the number of pathways. Clearly, if you increase the number of generations, only selecting one possible pathway, you dramatically decrease the total fraction of experiments you have to do. However, this doesn't address the possible drop in difference between the RSIs seen by generation 3, which, if I've understood correctly, may be a limiting factor in the number of generations/reaction steps you can perform. This also doesn't address whether selecting only one pathway is always the best option, or whether discarding potentially interesting alternatives might occur with this approach.

We thank the referee for this comment and for the opportunity to clarify this point. This methodology focuses on searching for the highest reactivity in the analysed mixtures rather than on the highest yields of a target compound. In the first generation, the reactivity levels of the four reaction mixtures are well distinct from each other. However, when the reactivity of the reaction mixtures become more and more comparable, we observe a drop in difference among the RSIs. This is also a valuable piece of information that can be extracted from this methodology, i.e. some regions of the reaction space give a large variation in reactivity, whereas others are really similar, thus indicating there is smaller difference in reactivity within this particular subset of the parameter space. Despite that, we demonstrate that our methodology remains very accurate in selecting the pathway with the highest reactivity. As previously mentioned, the reduction in sensitivity as generations proceed is due to a dilution of the overall reaction mixture. Here, we demonstrate that it is possible to accurately discriminate within 64 (4x4x4) possible pathways in three consecutive steps, which in our opinion proves the concept. As previously mentioned, there are techniques developed to concentrate the samples in-flow by either distillation or membrane separation, which can be applied to remove solvent from the reaction media, thus increasing the concentration. In this way, it would be possible to continue increasing the number of steps/generations.

Reviewer #2 (Remarks to the Author):

The paper describes a technically interesting system, but to me does not bring out the point of the work very well. (I enjoyed reading the supplementary more than the paper). The paper seems to be presented as a way to make novel compounds using several sequential reactions without intermediate work-up / purification (or at least to identify which of many possible products is from the most 'reactive' pathway).

We thank the referee for this comment and for the opportunity to clarify that our methodology is not a target oriented strategy. In fact, we are not claiming to have a new procedure to make complex molecules using multistep process without any work-up or purification steps. It has been now made more explicit that we present here a methodology to explore the chemical space by looking at reactive pathways rather than for a new procedure that looks at the optimal formation of specific outcomes in highest yields. We added the following text:

The idea is not just to follow yield, but to see how using a straightforward algorithm, following maximum reactivity, can lead to linking together many reactions with minimum work-up allowing the system to autonomously follow reactivity.

The paper uses the odd term 'Reactive' as it is essentially following loss of starting material, not formation of product (or products). For multi-step synthesis the interest is much more in selectivity (and consequently, yield). It might perhaps be more clearly presented as away to discover new reactivity (as it just looks a loss of starting materials) in which case the need for several steps needs to be justified. Perhaps the case can be made that the penultimate steps provide a more complex (and not commercially available) precursors for the last stage? The approach would be more convincing in a scenario where most of the reactions do not work.

We are not trying to make a target outcome or many intermediates that can then lead to a specific molecule. Instead the focus is on the exploration of the chemical space driven by the degree of reactivity in the pathway. If we remove the steps, we would limit the chemical space and we would not have any more a decision-making system. We then use also stoppers to check that the algorithm is correctly working, by focusing the chemical exploration on the most reactive regions.

The abstract is poor - more on an introduction. It needs to concisely state what the research has delivered.

We have totally rewritten the abstract to me more concise.

"However, the autonomous exploration of chemical space has been limited by the need for separate work-up-separation steps searching for molecules rather than reactivity." Which seem to be saying that looking for what is formed is not as good as looking at what has reacted - something I suspect most chemists would disagree with. The 'work-up-separation' is irrelevant - crude products can be analysed in many ways for selectivity in product formation.

We do agree, but here we present a new methodology to look for the region of the chemical space with the highest reactivity rather than yields, to enhance the possibility of discovery of new products. We are not suggesting a new approach to do chemistry, but a new approach to explore the chemical space.

"target-driven molecular synthesis, searching chemical databases" What is the relevance of the last part - how does it link to the first?

This was confused, we mean that automation normally makes pre-defined libraries of molecules. This is now corrected.

"Herein we present a reactivity explorer robot that is capable of navigating a large reaction network and assessing the reactivity of the chemical transformations autonomously, without needing to do every reaction." And later "autonomously assess, and rank the reactivity of all reagent combinations" It does not assess the reactivity of all the transformations - it just picks the best from the first step, and only tries the second step reagents on this one (so gets no information on the other possibilities). There is no reason why reactivity in the second step should be correlated to that in the first step (except when the first step has failed in a way that the functionality expected for the second step is not present). Better to say "attempts to find an overall sequence of reactions where every stage results in maximum loss of starting material (with the reasonable hope that it also maximize production of a particular product).

We thank the referee for this comment and as we have seen from reviewer 1 we acknowledge the need to clarify our message. It has been now made more explicit that we present a new methodology that works in an uneven space. In fact, we cut the chemical space after each step to finally focus on its overall most reactive region. Each step-selected contribute to the final selection as we are going step by step closer to the identification of the most reactive pathway in that highly reactive region of the space.

What does Figure 2 add - to me is just confuses as it seems to imply that many reaction paths have been examined). Fig 1 at least shows the pruning being done. The comparison to the combinatorial approach in Fig 1 might be more useful if it showed the reality - in such a combichem approach many of the routes would fail (possibly in the first step) so only a proportion (probably small) of the third generation

compounds would be as required. Method B might only make 4 compounds, but at least the chance of these 4 being correct is higher as the 'best' generation 1 and 2 reactions have been selected.

We disagree with the referee. Figure 1 helps to visualize the concept and approach we are presenting for the exploration of the chemical space reactivity-driven. In Figure 2 we are instead illustrating the actual chemical space/model system used to develop this methodology, by highlighting the pathway selected among all possibilities.

Technically the authors look at the least squares difference between reaction product and the spectra which would be expected from starting materials (including the crude products from previous steps). This only approximates to the loss of starting material as the extent to which the products change the region of the IR spectra observed matters as well (i.e. different products will have different effects so the RMS difference for the same extent of reaction will be different in different cases). For example I suspect that all three acid chlorides in the last step reacted to completion - so the difference in RSI is due to the differing products not differing reactivity's. Using GC or UPLC to monitor disappearance of starting material would be more reliable and since only a single data point is taken, not significantly more challenging.

We thank the reviewer for these thoughtful insights. We think that IR spectroscopy is a more robust technique than GC or UPLC to look at the reactivity of organic molecules. IR spectroscopy is a versatile and universal technique, that allows to follow the changes occurring in a reaction mixture in real-time. Contrary to GC and UPLC, IR spectroscopy does not require to have knowledge of the final product or target molecule in the mixture in order to develop a suitable detection method. The reviewer is correct in his interpretation of the chemistry. However, the main objective of our work is to develop a method that explore complex chemical landscapes autonomously, i.e. with minimal (or none at all) human supervision. Hence, the method should be applicable to all types of chemistries without biasing them with pre-existing knowledge. In this sense, we believe that the methodology is successfully demonstrating that there is a reasonably similar RSIs, which indicates similar reactivities in this step, in agreement with the reviewer observation. Nevertheless, the system has reached this conclusion based on empirical unbiased data. In the future we can extend this, but it is important to prove the principle.

The 'validation' (Table 1) uses the very special case where a strongly IR absorbing group (a carbonyl) shifts between starting material and product. It also uses the same reaction under different conditions so the same products are presumably being formed, just to a different extent- quite different to the main topic of the paper where different products are being formed. I do not object to the technique, but the problems need to be acknowledged.

We agree and also acknowledge this point. We have now made the methodology chosen here more explicit that i.e. it looks for the highest reactive pathway that in this case, where the complexity of the reaction mixture is contained, also corresponds to the product formed in highest yields. However, we further demonstrate that this methodology is giving a good approximation of which pathway will yield the highest reactivity of the starting materials, rather than the overall highest yields among the possible 64 outcomes.

Needs to be careful with the spelling of chemical names: methacroleine (2a), trans-2-methyl-pent-2-ene

We apologize for this misspelling; this has now been corrected.

References 15 and 20 are the same as are 16 and 21!

We apologize for these repetitions; these have now been corrected.

Overall I feel that the work is well worth publishing, but with most of the hype taken out and consequently a much clearer description of what has been achieved.

We have rewritten the title, abstract and also addressed the introductory paragraphs to help make the work clearer and to remove any hype.

Reviewer #3 (Remarks to the Author):

Review of NCOMMS-17-00922

This is a very interesting and novel paper. I think that it is definitely publishable and potentially suitable for Nature Communications. The concept of using automated reactors in this way is very promising and clearly has wider applications beyond this demonstration of concept. I have two serious points that need to be addressed.

1. There is no description or diagram of the equipment which would enable anyone else to replicate the work. Figure S1 is a not particularly good photo which does little more than give an impression of the equipment. What is needed is a clear piping diagram with a list of components/suppliers, showing how everything is interconnected so that others could repeat the work if they wished.

This is a good point. To address this we made changes in the SI by replacing Figure S1 with a schematic representation of the flow setup.

2. I feel that there is too much hype in the title “An autonomous organic reaction robot that searches for chemical reactivity”. Clearly, they are using quite a complex set of pumps but I don’t think that one is justified in describing such a set up as a “robot”.

We have changed the title. We present a liquid-handling system that can be programmed to autonomously carry out a sequence of complex operations (reaction synthesis, real-time analysis, and data processing). In the strict sense of the term, it is a robot, but to make the things more clear we have changed the title to “**An autonomous organic reaction search engine for chemical reactivity**”

Similarly the algorithms, although clever, are not really artificial intelligence in the sense that I understand.

We agree. The barrier between AI and algorithms, especially with feedback control, is small but the use of the word algorithm is more accessible and it is important to be clear since we are aiming to influence chemists to think differently. We have therefore removed the term and replaced it with ‘algorithm’.

Also, the statement on page 2, line 8/9 about “the discovery of new reactions and molecules is still firmly in the domain of the chemist” is perhaps over dogmatic. For example, the paper “Automated Serendipity with Self-Optimizing Continuous-Flow Reactors” by Amara et al. EUROPEAN JOURNAL OF ORGANIC CHEMISTRY, 2015, 6141, DOI: 10.1002/ejoc.201500980, describes the discovery of a previously unknown reaction.

We thank the reviewer for this suggestion; we have now included this reference in the manuscript and toned down the statement to reflect this.

In summary, this Communication presents an excellent piece of work that needs to be described more fully in the Supplementary Information and is good enough to stand by itself without the inflated claims in the title and some of the terminology.

(A minor point: The use if the abbreviations ESI, RSI and SI all in the same paper is slightly confusing).

We have now changed SI into supporting information.

REVIEWERS' COMMENTS:

Reviewer #1 (Remarks to the Author):

I thank the authors for their responses to my comments, which have generally satisfied the points raised.

The addition of the following paragraph:

"This methodology could be further expanded to become a discovery tool but in future we will need to evaluate this approach next to more traditional approaches to see if discoveries can be made more quickly, and be more 'exciting' or unpredictable if reactivity is followed rather than explicit target based searching"

does not, to my mind, discuss the possible drawbacks of the method, so I still feel this is lacking from the paper. I understand that searching for reactivity is the job of the reported system and why this is an advantageous method for automation, but without a comparison to current methodologies the argument for its use is less convincing than it could have been. However, I don't think this is sufficient grounds for rejecting publication.

I also welcome the discussion of dilution as the reason behind the decreasing range of RSIs but would have liked to see more of the discussion in the rebuttal being included in the paper (for example, the strategies that could be used to alleviate this issue).

Beyond this the points I raised have been satisfactorily answered so I support publication in Nature Communications.

Reviewer #2 (Remarks to the Author):

The authors have written a good rebuttal to the referees comments, though perhaps less has made its way through to the paper.

The key idea / advance is that by comparing the combined IR signature of the starting materials with that obtained after mixing / reacting the machine can autonomously detect if something has happened.

I still feel that this is rather buried in the paper – most will misinterpret it as being about automated synthesis of molecules.

However, the changes have helped, and overall it is worth publishing with only minor changes – leave the reader to make up their own mind.

"We show the RSI correlates with reactivity "

is only true for the special case of the optimisation of step 1 (and as the paper is not about reaction optimisation this has limited relevance). For the rest it is good at picking up non-reaction, but otherwise is more a reflection of the differences in IR of SM and products.

Instead of "rather than relying on databases", perhaps just "rather than relying on prior knowledge"

"The idea is not just to follow yield.."

should be deleted – the work does not follow yield at all, it does not even care what product is formed.

Reviewer #3 (Remarks to the Author):

As Reviewer No 3, I am now happy that the MS has been modified in accordance with my comments. Specifically, most of the hype has been toned down and the Supplementary Information now contains much more specific experimental detail. I would be happy for it now to be published.

Referee replies in bold

Reviewer 1

1.The addition of the following paragraph:

"This methodology could be further expanded to become a discovery tool but in future we will need to evaluate this approach next to more traditional approaches to see if discoveries can be made more quickly, and be more 'exciting' or unpredictable if reactivity is followed rather than explicit target based searching" does not, to my mind, discuss the possible drawbacks of the method, so I still feel this is lacking from the paper. I understand that searching for reactivity is the job of the reported system and why this is an advantageous method for automation, but without a comparison to current methodologies the argument for its use is less convincing than it could have been. However, I don't think this is sufficient grounds for rejecting publication.

We have added an additional comment but we don't agree with all this comment.

I also welcome the discussion of dilution as the reason behind the decreasing range of RSIs but would have liked to see more of the discussion in the rebuttal being included in the paper (for example, the strategies that could be used to alleviate this issue).

We think this is not that much of a problem and it is more important to follow this up.

Reviewer 2

The authors have written a good rebuttal to the referees comments, though perhaps less has made its way through to the paper.

The key idea / advance is that by comparing the combined IR signature of the starting materials with that obtained after mixing / reacting the machine can autonomously detect if something has happened. I still feel that this is rather buried in the paper – most will misinterpret it as being about automated synthesis of molecules.

We have tried to make this more clear.

However, the changes have helped, and overall it is worth publishing with only minor changes – leave the reader to make up their own mind. "We show the RSI correlates with reactivity " is only true for the special case of the optimisation of step 1 (and as the paper is not about reaction optimisation this has limited relevance). For the rest it is good at picking up non-reaction, but otherwise is more a reflection of the differences in IR of SM and products. Instead of "rather than relying on databases", perhaps just "rather than relying on prior knowledge"

We have made this change.

"The idea is not just to follow yield.."

should be deleted – the work does not follow yield at all, it does not even care what product is formed.

We have made this change.

Reviewer #3 (Remarks to the Author):

As Reviewer No 3, I am now happy that the MS has been modified in accordance with my comments. Specifically, most of the hype has been toned down and the Supplementary Information now contains much more specific experimental detail. I would be happy for it now to be published.

We are grateful to the referees for their comments!